# Proposal: A Multi-Agent Framework for Reliable Drug-Target Interaction Prediction

**Bowen Gao 2024310685**
**Peidong Zhang 2024310681**
**Huajun Bai 2024310650**

## 1 Background

Predicting whether a drug (small molecule) can bind to a specific target (protein pocket) is a crucial task in the field of drug discovery. Accurate prediction of drug-target interactions (DTIs) can lead to the identification of potential new drugs for the treatment of diseases, thereby significantly benefiting human health. With the success of machine learning, numerous machine learning-based models have been developed to address this task.

Recently, the development of large language models (LLMs) has opened new avenues for solving complex scientific problems. Initial attempts have been made to apply LLMs to drug discovery tasks, including DTI prediction. However, leveraging LLMs in scientific domains presents unique challenges. One major issue is the phenomenon of AI hallucination, which is especially problematic in fields requiring specialized domain knowledge. Another challenge is the inconsistency of predictions, leading to unstable and unreliable results.

In this project, we aim to explore how to efficiently harness the power of LLMs in the DTI prediction task. By addressing the challenges associated with applying LLMs in this domain, we hope to develop a framework that can effectively utilize LLMs for accurate and reliable DTI prediction. Successfully solving this problem could accelerate the drug discovery process by identifying promising drug candidates more efficiently. Moreover, we anticipate that our proposed framework could be extended to other scientific discovery domains where naive application of LLMs is non-trivial.

## 2 Definition

In this proposal, we focus on the Drug Target Interation prediction task. Given the information about a protein target and a small molecule as context, with the goal of answering whether they can bind (binary classification) or assessing their binding affinity (regression). Notably, the term "agent" refers to the pre-trained general-purpose large language model.

## 3 Related Work

### 3.1 Drug-Target Interaction Prediction with LLM

Embeddings from specialized LLMs are commonly used for DTI prediction within specific domains Singh et al. [2023]. Additionally, general-purpose LLMs like Galactica Taylor et al. [2022], trained on scientific knowledge, can predict protein-drug interactions. While their application in drug discovery has been explored, there remains room for improvement.

### 3.2 Multi-Agent Collaboration

Recently, multi-agent frameworks based on LLMs have garnered significant interest in both industry and academia. Naively chaining LLMs often leads to cascading hallucinations. Advanced models like

Preprint. Under review.

MetaGPT Hong et al. [2023] have shown promising results in complex, cooperative AI applications. In the field of drug discovery, models like DrugAgent demonstrate the capability of large language models for reasoning and explainable AI in drug repurposing Inoue et al. [2024].

## 4 Proposed Method

We propose a domain-knowledge-guided, multi-agent collaboration framework to address the issues of inconsistency and hallucination that often arise when using a single LLM agent for drug-target interaction (DTI) prediction. Our motivation is inspired by how humans tackle complex real-world tasks: by breaking them down into several standard operating procedures, each defining a sub-task managed by specialized individuals or units. This approach allows each person to focus on a single, clearly defined, and manageable problem.

Predicting interactions between drugs and targets is inherently complex and requires expertise from various domains. Relying on a single LLM for such predictions can be overly challenging, leading to errors and hallucinations. To address this and mimic how real-world pharmaceutical experts approach complex problems, we decompose the overall task into several sub-tasks, including protein sequence analysis, drug molecule analysis, and binding analysis. Each agent is assigned a specific role to solve one of these sub-tasks, and their collective outputs are integrated to generate the final prediction.

To further enhance prediction consistency, we introduce a multi-agent, debate-based ensemble method. In this framework, referred to as a "group," multiple groups of agents perform predictions independently, forming a mixture-of-experts model based on LLM agents. If discrepancies arise among the predictions from different groups, a debate is initiated, where agents present their reasoning processes. A designated "judge" agent then evaluates these arguments and selects the most reliable prediction. This approach ensures that the final prediction is both well-reasoned and consistent across multiple runs. The proposed framework is shown in Figure 1.

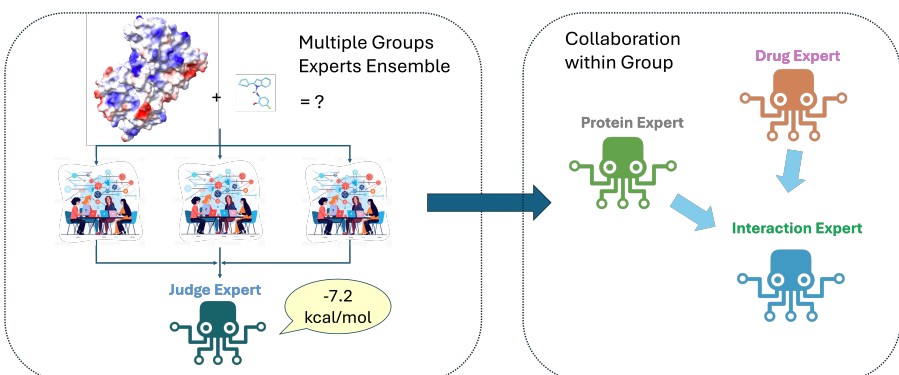

Figure 1: A multi-agent collaboration and mixture framework for reliable drug-target interaction prediction using LLM agents.

We use the BindingDB dataset Liu et al. [2007], a public database that contains measured binding affinities for protein–ligand interactions. This dataset includes essential labels such as dissociation constants (Kd), inhibition constants (Ki), and IC50 values, which are crucial for understanding molecular interactions in drug discovery.

We will test our framework and compare it with single large language models (LLMs) across different baseline models, including Llama-3 by at Meta AI [2024], GPT-4o by OpenAI [2024] and GLM-4-Plus by Zhipu AI [2024]. Additionally, we will compare our framework with state-of-the-art machine learning models that have been fine-tuned on these datasets and are expected to yield superior results.

Our objective is to determine whether our framework can enhance the performance of using only a single LLM for prediction and achieve results comparable to or better than those of fine-tuned machine learning models. Beyond accuracy metrics, we will also evaluate the consistency of our framework's predictions by performing multiple prediction runs and calculating the variance between them.

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
