# OpenReview forum: "【Proposal】A Multi-Agent Framework for Reliable Drug-Target Interaction Prediction"
_tsinghua.edu.cn/THU/2024/Fall/AML — THU 2024 Fall AML Submission_

### Official Review · ~Guangjie_Xu1 · 2024-11-09

**Rating:** 9
**Confidence:** 5

**Review:**

This proposal offers a promising approach to enhance the reliability of drug-target interaction (DTI) prediction through a multi-agent framework utilizing large language models (LLMs). Overall, the structure is complete and clear, and the tasks are well-described with logic and rigorous language.

**Pros**
1. Innovative Multi-Agent Framework: The framework’s approach of using multiple agents, each specializing in a sub-task, is a unique solution that mimics real-world problem-solving methods, making it highly suitable for complex DTI prediction tasks.

2. Improved Prediction Consistency: The structured “debate” process among agents and the role of a “judge” agent help ensure that the final prediction is consistent and well-reasoned, reducing potential prediction variability.

3. Potential for Broad Application: The framework’s design could be extended beyond DTI prediction, offering a structured and reliable method for scientific discovery tasks where traditional LLMs struggle with accuracy and stability.

**Cons**
1. High Computational Cost: The multi-agent approach, especially with ensemble and debate mechanisms, may require significant computational resources, which could impact feasibility in large-scale applications.

2. Scalability Concerns: Applying this framework to other domains may present scalability challenges, as it relies on multiple agents and complex decision processes.

---

### Official Review · ~Guanglei_He1 · 2024-11-10
**The problem definition is very clear, but it is difficult to truly harness the capabilities of LLMs. Ensuring fairness in horizontal comparisons between different LLMs is a significant challenge.**

**Rating:** 9
**Confidence:** 3

**Review:**

The overall proposal appears to be very concise, clearly articulating the background, definitions, challenges, existing methods, and objectives of the problem.

Although I am not deeply familiar with this field, I understand that accurately determining whether a drug (small molecule) can bind to a specific target (protein pocket) is a crucial task. This is a complex process that also requires experimental validation.

Therefore, I believe that utilizing existing confirmed data and merely employing Large Language Models (LLMs) for replication—to assess their potential in this domain—should not pose any issues.

The entire proposal seems to primarily leverage the capabilities of existing models. The design of prompts for different models has a decisive impact on the results; under such circumstances, maintaining fairness and impartiality is quite challenging. This aspect requires particular attention.

---

### Official Review · ~Ruitao_Jing1 · 2024-11-11
**A Creative Multi-Agent Framework**

**Rating:** 9
**Confidence:** 3

**Review:**

The paper presents a novel approach to drug target prediction using a multi-agent LLM, addressing a problem of significant practical relevance with clear definition and robust dataset and evaluation design. The clarity in problem definition and the innovative methodology are commendable, highlighting the paper's potential impact in the field.

The authors propose a three-step process for prediction: protein sequence prediction, drug molecule analysis, and drug-target binding analysis, with each step handled by a group of agents and a referee mechanism to ensure reliability and consistency. This multi-step, multi-agent approach is a creative solution that could enhance prediction accuracy.

However, the paper could benefit from a more explicit delineation of how the proposed method differs from and innovates upon previous work. Additionally, further elucidation on leveraging the dataset to train a group multi-agent framework akin to MoE would strengthen the paper's contribution to the literature.

---

### Official Review · ~Nan_Sun10 · 2024-11-11
**Innovative Multi-Agent Framework for Drug-Target Interaction Prediction**

**Rating:** 9
**Confidence:** 3

**Review:**

This work proposes a multi-agent framework leveraging large language models (LLMs) to predict drug-target interactions (DTIs).

The framework divides complex DTI prediction tasks among multiple LLM agents, assigning specialized roles to each agent. It also introduces a debate-based ensemble approach, which is innovative in ensuring consistent, well-reasoned outputs.

The paper holds potential as it addresses notable challenges such as AI hallucination and prediction inconsistency, common in scientific applications of LLMs.

The authors aim to validate this framework using the BindingDB dataset, comparing it with state-of-the-art LLMs and machine learning models fine-tuned on similar tasks.

However, the use of multiple agents and debate mechanisms may lead to high computational requirements, limiting practical scalability. And an assessment of the limitations, such as handling large datasets or scalability issues, would strengthen the paper.

---

### Official Review · ~Chaoqun_Yang2 · 2024-11-12
**Novel and interesting idea**

**Rating:** 9
**Confidence:** 4

**Review:**

**Summary:**
The paper proposes a multi-agent framework that leverages large language models (LLMs) to predict drug-target interactions (DTIs) with a focus on addressing the issues of AI hallucination and prediction inconsistency.  The framework decomposes the DTI prediction task into sub-tasks managed by specialized agents and employs a debate-based ensemble method to enhance prediction consistency.

**HighLights:**
The proposal presents an innovative approach to DTI prediction by integrating multi-agent systems with LLMs.  The idea of using specialized agents for sub-tasks and a debate-based ensemble method to resolve inconsistencies is novel and has the potential to significantly improve the reliability of DTI predictions. The proposed framework could have a significant impact on the field of drug discovery by improving the accuracy and reliability of DTI predictions.  If successful, this approach could accelerate the drug discovery process and potentially be applied to other scientific domains.


**Advice:**
1.  **Comparison with Existing Methods:** A more detailed comparison with existing DTI prediction methods, including both LLM-based and non-LLM-based approaches, would provide a clearer understanding of the advantages of the proposed framework.
2.  **Limitations and Future Work:** The authors should discuss potential limitations of their approach and suggest directions for future research to address these limitations.
3.  **Ethical Considerations:** Given the application in drug discovery, the authors should consider including a section on the ethical implications of their work, particularly regarding the reliability and potential biases in AI predictions.

---

### Official Review · ~Xuancheng_Li1 · 2024-11-12

**Rating:** 10
**Confidence:** 4

**Review:**

Summary
This proposal introduces a multi-agent framework leveraging large language models (LLMs) to predict drug-target interactions (DTIs), a critical step in drug discovery. Recognizing the challenges of using single LLMs, such as AI hallucinations and prediction inconsistencies, the authors propose decomposing the task into specialized sub-tasks handled by different agents. Each agent tackles distinct aspects like protein sequence or drug molecule analysis, and results are aggregated for a final prediction. To further ensure reliability, multiple agent groups debate discrepancies, with a "judge" agent determining the final consensus.

Strengths
The framework is innovative, using a multi-agent setup and debate mechanism to improve consistency and accuracy in DTI prediction. This approach mirrors expert collaboration in complex tasks, potentially enhancing the robustness of LLM predictions in scientific applications.

Weaknesses
The framework’s feasibility and scalability in large datasets or multi-drug scenarios remain unclear. Additional details on how agents’ sub-task outputs are integrated and validated would improve clarity.

Conclusion
This project offers a creative approach to enhance DTI prediction, with promising implications for AI-driven drug discovery. Testing the framework across various LLMs and traditional models, as planned, will reveal its effectiveness and potential applications beyond drug discovery.

---

### Official Review · ~Justinas_Jučas3 · 2024-11-12
**Innovative Idea and Very Clear Proposal with Clear Disadvantages**

**Rating:** 7
**Confidence:** 4

**Review:**

A very well-structured and well-written (in terms of simplicity of reading) report. The problem of detecting drugs is very relevant. However, the proposal contains quite some unclarities regarding the logic.

## Advantages
1. Well-structured, almost all requirements are satisfied
2. Very original idea and a unique, and very clear proposed solution
3. Some visualizations are added, that make the reading more clear

## Disadvantages
1. I cannot find any evaluation metrics on how the performance of you approach will be evaluated in comparison of other algorithms (this was a requirement!)
2. For me, at least a short explanation is lacking, why a LLM approach is chosen over some other traditional method, such as using simple neural networks or other classifiers. Perhaps I am wrong, but it seems like an overkill for the problem. Is it because of explainability?
3. Personal, it is not really clear what is meant by a 'debate' between LLM agents, in case of discrepancies. This part is not elaborated in the report.
4. The realistic possibility of actually completing this assignment within a month sounds rather unrealistic given the time constraint.
5. The definition of the problem is not clear (at least the second sentence, which contains all of the needed logic). Could be elaborated.

---

### Official Review · ~Yangchi_Gao1 · 2024-11-12
**innovative idea**

**Rating:** 9
**Confidence:** 4

**Review:**

The proposal proposes a promising multi-agent framework that could significantly advance DTI prediction. Its innovative approach to leveraging LLMs and addressing the challenges of AI hallucination and prediction inconsistency is commendable.

Advantage:
1.Multi-agent collaboration: The framework's use of multiple agents for different sub-tasks reflects a human-like approach to problem-solving, potentially leading to more accurate and reliable predictions.
2.Debate-based ensemble method: The introduction of a "group" of agents that debate and justify their predictions before a "judge" agent selects the most reliable one is an innovative method to ensure consistency and reduce errors.

Disadvantage:
1.There is a risk of overfitting the model to the training data, especially if the dataset is not diverse enough.

---

### Official Review · ~Fei_Long3 · 2024-11-12
**A novel and promissing idea in DTI prediction**

**Rating:** 9
**Confidence:** 4

**Review:**

The proposal presents a promising approach to DTI prediction using a multi-agent framework.

**Strengths**:

**Novel Framework**: The proposal introduces a multi-agent framework that leverages the power of LLMs for DTI prediction, which is a significant step forward in the field of drug discovery.

**Decomposition of Tasks**: The approach of decomposing the DTI prediction task into sub-tasks is well-justified and aligns with how experts in the field approach complex problems, which could lead to more accurate and focused predictions.

**Multi-Agent Debate Mechanism**: The introduction of a debate-based ensemble method to resolve discrepancies among predictions is an innovative approach to improving prediction consistency and reliability.

**Weakness**:

**Evaluation Metrics**: The evaluation plan focuses on accuracy and consistency but does not mention other important metrics, which are also critical in assessing the performance of prediction models.